# Anti-Viral Pattern Recognition Receptors as Therapeutic Targets

**DOI:** 10.3390/cells10092258

**Published:** 2021-08-31

**Authors:** Conor Hennessy, Declan P. McKernan

**Affiliations:** Pharmacology & Therapeutics, School of Medicine, Human Biology Building, National University of Ireland Galway, H91 TK33 Galway, Ireland; conor.hennessy@stcatz.ox.ac.uk

**Keywords:** pattern recognition receptors, toll-like receptors, RIG-like receptors, viral infection, anti-viral drugs, vaccine adjuvant

## Abstract

Pattern recognition receptors (PRRs) play a central role in the inflammation that ensues following microbial infection by their recognition of molecular patterns present in invading microorganisms but also following tissue damage by recognising molecules released during disease states. Such receptors are expressed in a variety of cells and in various compartments of these cells. PRR binding of molecular patterns results in an intracellular signalling cascade and the eventual activation of transcription factors and the release of cytokines, chemokines, and vasoactive molecules. PRRs and their accessory molecules are subject to tight regulation in these cells so as to not overreact or react in unnecessary circumstances. They are also key to reacting to infection and in stimulating the immune system when needed. Therefore, targeting PRRs offers a potential therapeutic approach for chronic inflammatory disease, infections and as vaccine adjuvants. In this review, the current knowledge on anti-viral PRRs and their signalling pathways is reviewed. Finally, compounds that target PRRs and that have been tested in clinical trials for chronic infections and as adjuvants in vaccine trials are discussed.

## 1. Introduction

Infection as well as tissue injury/stress can activate inflammatory responses. Inflammation is widely recognised by the cardinal symptoms of fever, redness, oedema, pain, and loss of function. Inflammation is necessary to help with the removal of the source of infection, to help protect the infected tissue(s), and to restore homeostasis. Both immune and non-immune cells produce cytokines, chemokines, and vasoactive peptides. These molecules attract immune cells such as neutrophils that are normally restricted to the vasculature, allowing them to enter into the inflamed or infected tissue [1,2,3]. Viruses cannot survive or replicate themselves and so need a host. They possess several distinct features that have allowed our immune system to develop a number of strategies to detect and remove viruses [4]. However, viruses have in turn developed strategies of their own to hide and evade our immune system. They can be detected at several stages in their life cycle and our immune system usually initiates an appropriate response [5]. These responses are usually initiated by a key family of receptors known as pattern recognition receptors (PRRs) that recognise pattern-associated molecular patterns (PAMPs). The binding of PAMPs to PRRs results in the initiation of an appropriate and regulated inflammatory response to the infection [6]. PRRs can be grouped into RIG-like receptors (RLRs), NOD-like receptors (NLRs), C-type lectin receptors, and Toll-like receptors (TLRs). Stimulating these receptors may help in the fight against infection. This review will discuss the current knowledge specifically on anti-viral PRRs and molecules that target them that have been tested to date in clinical trials.

## 2. Anti-Viral Pattern Recognition Receptors

Viruses can be categorised as RNA viruses, DNA viruses or retroviruses (which can have RNA or DNA in some cases). They can also be categorised by whether their nucleic acid is positive or negative sense. In addition, replication of these viruses also generates intermediate molecules such as ssRNA and/or dsRNA [7,8]. These nucleic acids are distinct from the host and are often contained in a specific compartment following infection separate from host nucleic acid [9,10]. Anti-viral pattern recognition receptors employ a variety of strategies to detect nucleic acids from viruses and some bacteria. Detection can occur in the endosomes (usually following phagocytosis by immune cells) or in the cytosol of immune and non-immune cells where they sense DNA or RNA from viruses that have shed their protein coat prior to replication or genome integration [11].

The first PRRs to be discovered were Toll-like receptors. They are most highly expressed in immune cells, but some are also expressed in non-immune cells such as epithelial cells. Basal expression levels vary across the body [12,13]. TLRs 1, 2, 4, 5, 6 and 10 are found on the cell surface and generally detect bacterial or fungal PAMPs and some viral proteins, whereas TLRs 3, 7, 8, 9 are located intracellularly and mostly detect viral and bacterial nucleic acids. The cellular localisation of PRRs is important to their function. Intracellular TLRs act within the endosomal compartment, which generally excludes host nucleic acid and allows them to better distinguish foreign RNA/DNA from that of the host, the in [14].

Other receptor families such as RLRs, NLRs and DNA sensors such as absent in melanoma 2 (AIM2) and cyclic GMP-AMP synthase (cGAS) can recognise a range of microbial nucleic acids in the cytosol [4]. This strategy allows the innate immune system to detect nucleic acids from viruses at different stages of their life cycle and cellular localisation. PRR activation culminates in the release of type I interferons and cytokines such as interleukin 1β (IL1β), IL6 and tumour necrosis factor α (TNFα) [11,15]. Type I interferon release acts in paracrine and autocrine pathways via interferon alpha receptor 1 and 2 (IFANR1/2). This leads to activation of signalling pathways that culminates in the induction and release of interferon stimulated genes (ISGs) such as 2′, 5′-oligoadenylate synthetase (OAS) and protein kinase R (PKR) that help suppress virus replication and assembly in the host cell [16]. Cytokines alter the vasculature to help activate and recruit immune cells to the site of infection [17].

In humans, there are 10 TLR family members that are membrane-bound receptors located intracellularly and on the cell surface [14]. Ligand binding is carried out by the extracellular domain, which consists of leucine-rich repeats (LRRs), while intracellular signalling is carried out by the intracellular Toll-interleukin-1 resistance domain (TIR) [18]. Ligand recognition initiates a signal transduction cascade within the cytoplasm via myeloid differentiation primary response gene 88 (MyD88), an adaptor molecule, and specific kinases such as IL1 receptor-associated kinases (IRAK) 1/2/4. A series of phosphorylation reactions activates transcription factors such as nuclear factor kappa-light-chain-enhancer of activated B cells (NF-κB), or interferon regulatory factors (IRF) 3/5/7. These then move to the nucleus inducing transcription of cytokines and interferons (Figure 1) [19].

The intracellular TLR, TLR3 was identified as recognising polyinosinic-polycytidylic acid (poly(I:C)), a synthetic analogue of double stranded RNA (dsRNA) [20]. TLR3 is important in recognising ssRNA viruses such as respiratory syncytial virus (RSV), encephalomyocarditis virus, West Nile virus and small interfering RNAs [19]. Its role has been illustrated using *tlr3* deficient mice, which are susceptible to lethal viral infection [21]. TLR7 recognises ssRNA from viruses such as vesicular stomatitis virus (VSV), influenza type A (IAV) and human immunodeficiency virus (HIV) but was originally described as recognising imidazoquinolinone chemical derivatives such as imiquimod and resiquimod [19,22].

Expression of TLR7 is relatively high in plasmacytoid dendritic cells (pDCs). TLR8 also recognises ssRNA and is similar phylogenetically to TLR7. Interestingly, no release of interferon or cytokines occurs in the absence of TLR7 following ssRNA stimulation but immune responses are normal in TLR8 deficient mice [19,22]. It may have other roles in immune response. TLR9 has been shown to recognise unmethylated CpG containing ssDNA and it induces the expression and release of Type 1 interferon [23]. Like TLR7, TLR9 has relatively high constitutive expression in pDCs and B cells. It is important in the prevention of replication by certain viruses such as herpes simplex virus (HSV) and adenovirus [24,25]. Further details of TLR signalling have been discussed in detail elsewhere [26].

TLR recognition of RNA and DNA is limited to the endosomal compartment of immune cells. Other PRRs have evolved for sensing nucleic acids in different cellular locations and cell types to detect viruses at different stages of their life cycle [27]. The second category of PRR are the cytosolic RNA sensors. These are DExD/H box helicases (DHX) and include retinoic acid inducible gene I (RIG-I), melanoma differentiation-associated antigen 5 (MDA5) and laboratory of genetics and physiology 2 (LGP2) which does not have the N-terminal caspase recruitment (CARD) domain. RIG-I and MDA5 are primarily involved in the cytosolic recognition of RNA viruses whereas LGP2 regulates MDA5 and RIG-I signalling [28]. RLRs are expressed across many tissues and cell types. RIG-I recognises short RNA molecules (<300 bp) in the cytosol containing 5′-diphosphates as well as 5′-triphosphorylated uncapped RNA. MDA5 recognises long dsRNA (>300 bp) as well as high molecular weight branched RNAs [7]. Both receptors use mitochondrial anti-viral signalling protein (MAVS) to signal leading to activation of the inhibitor of κB kinase (IKK)-related kinases TANK-binding kinase (TBK1) and IKKi which then activate the transcription factors IRF3 and IRF7 and induce type I interferon transcription (Figure 1) [15,29]. Viruses commonly recognised by RIG-I include (−)ssRNA viruses such as influenza A/B as well as (+)ssRNA viruses, e.g., hepatitis C virus whereas MDA5 commonly recognises dsRNA viruses as well as some (+)ssRNA viruses [4].

The third category of PRRs are cytosolic receptors that recognise microbial DNA during infection. DNA is mostly contained within the nucleus or mitochondria in mammalian cells. During infection, viruses inject DNA into the cytosol of cells. DNA sensors include absent in melanoma 2 (AIM2) and cyclic GMP-AMP (cGAMP) synthase (cGAS) among others. This then activates an endoplasmic reticulum-bound adaptor called stimulator of interferon genes (STING). STING undergoes a conformational change and inducing the production of type I interferon via the activation of TBK1 and IRF3 (Figure 1). All of these are widely expressed and recognise DNA from DNA viruses, retroviruses, genomic DNA and cyclic dinucleotides (CDNs) from bacteria and self-DNA from dead cells [30,31]. Recognition of cytosolic DNA by AIM2 activates the inflammasome leading to activation of caspase 1 which subsequently cleaves gasdermin and leads to the release of interleukin 1 and 18 [32,33]. Upon recognition of DNA of various sizes by cGAS it can synthesise cGAMP from ATP and GTP ([34,35,36].

## 3. Pharmacological Targeting of PRRs

PRRs and their signalling pathways could be pharmacologically targeted at the receptor level, the downstream kinases, or the protein–protein interactions between adaptor molecules. This review will highlight the receptors specifically as targets rather than downstream events. To date, PRR agonists, antagonists and antagonistic antibodies have been designed. Clinical trials have evaluated these molecules as adjuvants for vaccines and cancer as well as for infections, acute and chronic inflammatory diseases, and neurological conditions. This review will focus on those tested for anti-virals and adjuvants for vaccines. Reviews focusing on PRRs as targets in cancer and other inflammatory diseases can be found elsewhere [37,38].

### 3.1. PRR Agonists as Anti-Viral Drugs

Despite vaccination programmes being in place in many countries, there is a still an unmet clinical need to treat hepatitis B and C viruses, human immunodeficiency virus (HIV), human papillomavirus (HPV), herpes simplex virus (HSV) and more recently coronavirus (SARS-CoV-2) infections. These infections are associated with chronic disease and are lethal in many cases. Stimulating anti-viral pathways by using agonists of PRRs may help reduce the viral load and hence the associated disease. Hepatitis B virus (HBV) is a non-cytopathic DNA virus that is a common cause of liver disease. The virus generates intermediates of closed circular DNA in the nucleus of hepatocytes which have a long half-life and that get transcribed [39]. Stimulating the anti-viral immune response could aid in reducing chronic infection. As agonists of the anti-viral TLRs can induce anti-viral gene expression, these have been developed and tested in patients with chronic HBV infection but results to date have varied.

The TLR7 agonists tested to date in phase I trials include GS-9620, RO7020531 and TQ-A3334 were all shown to be safe and well tolerated in hepatitis B patients and healthy volunteers [40,41,42,43]. Vesatolimod (GS-9620) and TQ-A334 were also shown to induce anti-viral genes (e.g., ISG15) in a dose dependent manner [40,41,42]. However, in a phase II trial of GS-9260, there was no significant reduction in the levels of the disease marker hepatitis B surface antigen [40,42]. It has been suggested from ex vivo studies that TLR7 agonists may increase T and NK cell activity [44]. There have been many other trials conducted on these molecules (see Table 1) but results have not been reported. Due to their similar responses, TLR8 agonists such as selgantolimod (GS-9688) has been tested in healthy volunteers and chronic hepatitis B patients with no serious adverse events reported in either cohort as well as dose-dependent induction of cytokines such as IL-12 [45,46]. Finally, recent research has focused on the role the non-endosomal pattern recognition receptors, in particular RIG-I given the role of cytoplasmic nucleic acid sensors in mediating anti-viral responses. Based on the validation of its anti-viral activity in an animal model of chronic hepatitis B, an orally available prodrug (small molecule nucleic acid) and RIG-I/NOD 2 agonist called Inarigivir (SB9200) has recently been tested [47,48]. In phase II trials (ACHIEVE trial), it caused a reduction in HBV DNA and RNA and had mild/moderate adverse events reported [49,50,51]. There have, however, been a number of other trials with this molecule that have since been terminated due to reports of liver injury.

Chronic hepatitis C viral (HCV) infection can lead to liver cirrhosis and carcinoma [75]. Similar to hepatitis B infection, Toll-like receptor agonists have been tested but are at an earlier stage of development [76]. In phase I trials, the TLR7 agonists ANA773, GS-9620, RO7020531 and PF-04878691 have all been administered (mostly by oral route but also by other routes) to hepatitis C patients and healthy volunteers with no serious adverse events reported [52,53,54,55,56]. Significant decreases in serum HCV RNA levels in the highest dose group were reported as well as dose-related interferon responses [52]. Similarly, the TLR7/8 agonist Resiquimod (R-848) was shown to also reduce viral levels in HCV patients in a phase II trial but was associated with serious adverse events such as fever and lymphopenia [57]. TLR9 agonists including IMO-2125, CpG10101 and SD-101 have also been tested in phase I trials. These were safe, well tolerated and showed dose-dependent increases in cytokine levels and a reduction in HCV RNA when given subcutaneously for 4 weeks [52,53,54,55,56,57,58,59,60]. There have not been any further reports on testing and the extent of their efficacy is not known in these patients. Finally, Inarigivir (SB9200), the RIG-I/NOD 2 agonist, tested in phase I trials showed good activity against resistant hepatitis C variants after phase I testing [61]. It was also demonstrated to be safe, well tolerated and reduced viral replication A in randomised phase I ascending dose trial in chronic hepatitis C patients [60].

HIV infection affects millions worldwide and can become latent even in people who have been successfully treated with highly active anti-retroviral therapy (HAART). Viral particles can re-emerge later when therapy is interrupted. Using TLR agonists could be useful as latency reverting agents [77]. Given their ability to induce type I interferons and hence anti-viral genes, TLR3, TLR7 and TLR9, agonists have been tested in this domain. The TLR3 agonists Pol ICLC and Poly I:PolyC12U (Rintatolimod) were tested in phase I and II trials. Both were shown to have minimal toxicity. No significant effects on HIV levels were reported but an increase in CD4+ cell counts in patients and in the case of Rintatolimod could reduce the likelihood of category C HIV infection [62,63,64,65]. The oral TLR7 agonist GS-920 given to HIV patients on antiretrovirals showed an induction of immune cell activation and a decrease in proviral DNA in phase Ib trials but larger studies will be needed to confirm its efficacy [66,67]. In a phase Ib/IIa open label trial, the TLR9 agonist MGN1703 (Lefitolimod) led to activation of plasmacytoid dendritic cells, cytotoxic natural killer cells and CD8+ T cells as well as increased detection of HIV1 RNA in the plasma of a proportion of HIV patients when given subcutaneously. Separately, it was also shown to be safe and induce increased HIV1 T cell specific responses [68,69].

Human papillomavirus (HPV) infection is associated with the presentation of anogenital warts as well as cervical and anal cancers. Like many viral infections mentioned so far, imiquimod has been tested in trials for HPV. Administered to HPV patients with warts as a cream (5%) caused decreases in viral load as measured by HPV DNA as well as increases in IFN, TNF and IL12 mRNA [78]. HIV patients have a higher risk of anogenital warts and those patients on HAART were tested in an open label phase 4 trial where it was shown to be safe. Total wart clearance and decreasing HPV viral load was reported in a significant number of patients [70]. In subsequent double-blind RCT trials, those treated with imiquimod showed clearance for HPV6 and a significant decline in HPV11 viral load [71].

Herpes simplex virus II infection is also known to lead to genital warts, and it can establish persistent infection by evading immune responses [79]. Topical application of TLR agonists may therefore be of use in inducing an anti-viral response. The TLR7/8 agonist resiquimod has been tested as a topical gel applied twice weekly in phase II/III trials with varying results related to viral shedding and healing with adverse events such as erythema and erosion at the site more common in the treatment groups. Despite some efficacy in phase II trials, it was concluded that there was no efficacy with the treatment group in the phase III trials [72,73,74]. Finally, given the events of the past year, it is worth noting that several trials have begun investigating the effects of an aerosolised version of the mixed combination of TLR2/6 (Pam2) and TLR9 (ODN) agonists called PUL-042 in reducing the severity of COVID-19 caused by SARS-CoV-2 infection. No results were available at the time of writing.

### 3.2. PRR Agonists as Vaccine Adjuvant

For many years, vaccines have failed in trials due to a lack of efficacy and/or durability and often because immune responses were not strong enough to antigens. The use of adjuvants has helped resolve this but there was still a limited number of these molecules that were safe. PRR adjuvants have recently been test in trials due to their ability to bridge the innate and adaptive immune systems [80]. There have been several trials which have evaluated TLR agonists as adjuvants (see Table 2). Many of these have improved the efficacy and safety profiles of drugs and vaccines with the most effective being TLR3/5/7/8/9 agonists. In the case of Hepatitis B vaccines, they have been combined with TLR4, TLR7, TLR8 TLR9 agonists. Most of these trials reached phase III. The TLR4 agonists monophosphoryl lipid A (MPL) and its synthetic mimetic RC-529 have been combined separately with recombinant hepatitis protein vaccines and each were shown to be safe, well tolerated, and to induce higher sustained anti-HB titers as well as T cell responses years after administration [81,82,83,84]. The TLR7 agonist imiquimod and the TLR7/8 agonist resiquimod have also been combined with protein subunit vaccines for hepatitis B in separate phase II and phase III trials. Only results for imiquimod have been published and topical pretreatment with imiquimod cream at the injection site did not enhance the humoral response to 3 intradermal injections of the hepatitis B recombinant vaccine [85]. A combination of the TLR9 agonist CpG DNA and the hepatitis B surface antigen 1018 has been called Heplisav [86]. It was shown to be immunogenic and well tolerated in phase I trials and subsequently shown to significantly enhance humoral responses to the hepatitis B surface antigen vaccine [87,88]. This combination was demonstrated in phase III trials to improve immunogenicity and it allows for fewer doses over a shorter period [89,90,91].

Despite success with antiretroviral drug therapy, there is still a need for vaccines for HIV/AIDS. Many TLR agonists have been tested with various types of HIV vaccines. Poly ICLC, a TLR3 agonist has been combined with DCVax (a fusion protein consisting of an antibody to the dendritic cell receptor CD205 and the HIV gag p24 protein). This was tested in a phase I trial using healthy volunteers, but no results were available at the time of writing. The TLR4 agonist MPL and the NOD2 agonist muramyl dipeptide (MDP) were combined with the recombinant protein gp120 subunit vaccine in an initial phase I trial and showed neutralising antibody activity [92]. The TLR4 agonist RC-529 was combined with a multi-epitope peptide cytotoxic T lymphocyte HIV vaccine in initial phase I trials. While the vaccine was safe and tolerable it was only mildly immunogenic [93]. The TLR7/8 agonist 3M-052-AF and the TLR9 agonist CpG 1018 have been combined in separate trials with a gp140 vaccine in phase I trials and are still in the recruitment phase. CP7909, a TLR9 agonist was proposed to work as a latency inhibitor and in conjunction with antiretroviral therapy work to rescue the proviral reservoir in virologically suppressed infected patients [94].

Human papillomavirus vaccines have had success in recent years. Several trials have been designed to investigate the efficacy and safety of combining these with TLR agonists including MPL (TLR4) and imiquimod (TLR7). Cervarix, a vaccine against HPV16/18 strains uses an adjuvant called AS04, containing MPL and aluminium salt. In a phase III trial of over 2000 adolescent girls aged 10–14, the mean titers of antibodies to both strains were shown to be maintained at higher levels up to 10 years later in those vaccinated with the AS04 adjuvant [95,96]. TLR7 agonists have also been tested in trials with topical administration of imiquimod prior to vaccination with the HPV16 vaccine. It was reported in phase II trials that imiquimod treatment led to significantly greater local infiltration of T cells in lesion responders [97]. It has also been tested with the quadrivalent vaccine but did not show any differences in this instance [98,99]. There have been other phase II trials but either the results have not been reported or in one case, the trial was terminated due to lack of efficacy.

Herpes simplex virus 2 is regarded as a primary causative agent of genital herpes. Prophylactic vaccination is proposed to help reduce sexual transmission. An investigational HSV vaccine (glycoprotein D subunit vaccine) has been tested with some TLR agonists including MPL (as AS04) and the TLR9 agonist CpG1018. It was reported that the vaccine had an acceptable safety profile, it was well tolerated and immunogenic, as determined by geometric mean concentration of anti-gD2 antibodies. However, there was no control group to determine the effect of imiquimod [100].

Influenza infections cause thousands of deaths annually and mutation makes it difficult to control. Vaccination is key to reducing mortality. An intranasal quadrivalent flu vaccine FluMist was combined with the TLR3 agonist rintatolimod (Ampligen) and tested in a phase I/II trial. Despite being reported to be well tolerated, improving antibody titers, and demonstrating cross-reactivity to strains of avian flu, the trial was terminated [101]. Recently, a vaccine called VAX125 was generated by combining the TLR5 agonist flagellin with the H1N1 influenza virus HA1 domain. A phase I open label trial showed it to be safe and well tolerated when give as a single dose intramuscularly [102] and that it could induce a potent antibody immune response (10-fold over baseline) in elderly subjects [103]. Different domains of flagellin have been fused to the vaccine generating VAX128A, B and C. When tested in healthy subjects these were demonstrated to be safe and immunogenic [108]. VAX102 is a fusion of flagellin with the matrix protein 2 ectodomain. Phase I trial results demonstrated safety and immunogenicity [105]. A follow up study reported higher antibody levels in response to M2E (four-fold over baseline) [104].

Finally, imiquimod was given topically prior to an intradermal trivalent influenza vaccine and found to be safe, well tolerated. It also significantly increased, prolonged, and expedited the immunogenicity of the vaccine [106,107]. A phase III trial with young healthy volunteers confirmed as much as well as showing increased immunogenicity against non-vaccine strains where there is often antigenic drift such as H3N2 [106]. A phase I trial was conducted involving the topical administration of a resiquimod (TLR7/8 agonist) gel prior to the intradermal Intanza vaccine in elderly patients but results were not available at the time of writing.

## 4. Perspectives/Conclusions

In summary, agonists of ant-viral TLRs (TLRs 3, 7, 8 and 9) seem to hold the most promise as both anti-viral drugs and as vaccine adjuvants. As many of the PRR signalling pathways culminate in NF-κB and IRF activation (Figure 1), targeting the pathway downstream could affect all innate immune responses during infection. Therefore, it is more prudent to target upstream events which enhance or reduce specific PRR activity but allow other receptors to recognise and response to microbes. Inhibiting downstream PRR signalling could interfere with the cross-talk between individual pathways. Future approaches could be targeting specific domains in PRRs such as the LRR domain involved in ligand recognition in TLRs, targeting the ATP-binding sites or the NOD oligomerisation domain/NACHT domain in NLRs, targeting the TIR domain or recruitment of adaptors such as Mal or RIP2, targeting kinase activity or targeting some of the regulatory proteins involved in signalling. To date, most compounds evaluated in clinical trials are agonists of TLRs but that is not surprising given TLRs were the first PRRs to be described. As the importance of other PRRs in viral infections are uncovered it is expected that newer molecules will be developed and tested in trials. Overall, these receptors still hold a lot of promise as targets in inflammatory and infectious diseases.

## Figures and Tables

**Figure 1 cells-10-02258-f001:**
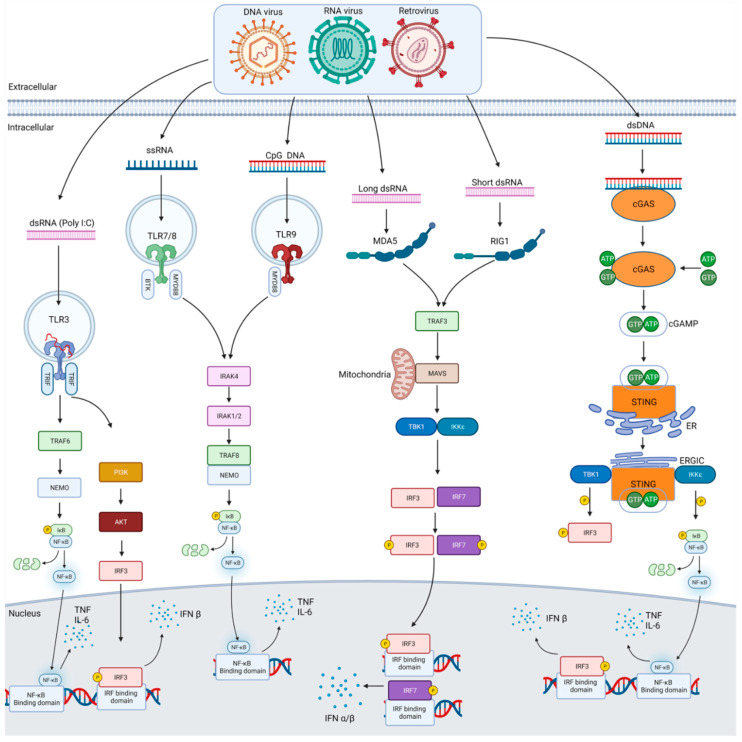
Anti-viral pattern recognition signalling pathways. Shown in this figure are the various signalling pathways used to respond to viral infection from DNA, RNA and retroviruses. Each of the major forms of nucleic acids that act as PAMPs from such viruses are illustrated along with the PRRs they bind to including TLRs, RLRs and DNA sensors such as cGAS. Each pathway activates specific kinases via their own adaptor molecules, and this ultimately leads to the activation of transcription factors that translocate to the nucleus and help initiate the transcription of cytokines such TNFα and IL6 or type I interferons, e.g., IFNβ, which then get released from the cell.

**Table 1 cells-10-02258-t001:** Shown in the table is a list of PPR agonists that have been tested in clinical trials for viral infections. Indicated is the viral infection, the name of the molecule, its target and the clinical trial information such as the trial number, the phase it has reached and the current status. HBV—hepatitis B virus; HCV—hepatitis C virus; HIV—human immunodeficiency virus; HPV—human papillomavirus; HSV—herpes simplex virus; SARS-CoV-2—severe acute respiratory syndrome coronavirus 2.

Virus	Molecule	Type	Target	Clinical Trial No.	Phase	Status	Reference
*HBV*	GS-9620	Agonist	TLR7	NCT02166047	II	Completed	[40,42]
	RO7020531	Agonist	TLR7	NCT03530917	I	Completed	[43]
				NCT04225715	II	Recruiting	
				NCT02956850	II	Active, not recruiting	
	TQ-A3334	Agonist	TLR7	CTR20182248	I	Completed	[41]
				NCT04180150	II	Recruiting	
				NCT04202653	II	Not yet recruiting	
	GS-9688	Agonist	TLR8	NCT03491553	II	Completed	[45,46]
				NCT03615066	II	Completed	
				NCT04891770	II	Not yet recruiting	
	SB9200	Agonist	RIG-I/NOD2	NCT03434353	II	Completed	[50,51]
				NCT03932513	II	Terminated	
				NCT04023721	II	Terminated	
				NCT04059198	II	Terminated	
				NCT02751996	II	Completed	
*HCV*	ANA773	Agonist	TLR7	NCT01211626	I	Completed	[52]
	RO7020531	Agonist	TLR7	NCT02956850	I	Active, not recruiting	
				NCT03530917	I	Completed	[53]
	PF-04878691/852A	Agonist	TLR7	NCT00810758	I	Completed	[54]
	GS-9620	Agonist	TLR7	NCT01591668	I	Completed	[55,56]
	Resiquimod	Agonist	TLR7/8		II		[57]
	IMO-2125	Agonist	TLR9	NCT00728936	I	Completed	[58]
				NCT00990938	I	Completed	
	CpG10101	Agonist	TLR9	NCT00277238	I	Completed	[59]
				NCT00142103	I	Completed	
	SD-101	Agonist	TLR9	NCT00823862	I	Completed	
	SB9200	Agonist	RIG-I/NOD2	NCT01803308	I	Completed	[60,61]
*HIV*	Poly ICLC	Agonist	TLR3	NCT02071095	I/II	Completed	[62]
	Rintatolimod	Agonist	TLR3				[63]
				NCT00000735	I	Completed	[64]
				NCT00001000	I	Completed	[65]
				NCT00035893	II	Completed	
				NCT00000713	I	Completed	
	GS-9620	Agonist	TLR7	NCT03060447	I	Completed	[66]
				NCT02858401	I	Completed	[67]
	MGN1703	Agonist	TLR9	NCT03837756	II	Recruiting	
				NCT02443935	I/II	Completed	[68,69]
*HPV*	Imiquimod	Agonist	TLR7	NCT00761371	IV	Completed	[70]
				CTRI/2009/091/000055	II/III	Completed	[71]
*HSV*	Resiquimod	Agonist	TLR7/8		II	Completed	[72]
					II	Completed	[73]
					III	Completed	[74]
*SARS-CoV-2*	PUL-042	Agonist	TLR2/6 and 9	NCT04312997	II	Active, not recruiting	
				NCT04313023	II	Recruiting	
				NCT02124278	I	Completed	

**Table 2 cells-10-02258-t002:** Shown in the table is a list of PPR agonists that have been tested in clinical trials as vaccine adjuvants. Indicated is the viral infection, the name of the molecule, its target and the clinical trial information such as the trial number, the phase it has reached and the current status. HBV—hepatitis B virus; HIV—human immunodeficiency virus; HPV—human papillomavirus; HSV—herpes simplex virus; IAV—influenza A virus.

Virus	Molecule	Type	Target	Clinical Trial No.	Phase	Status	Reference
*HBV*	MPL	Agonist	TLR4	NCT00698087	III	Completed	[81]
				NCT00697242	III	Completed	
				NCT02153320	III	Completed	[83]
					I	Completed	[84]
	RC-529	Agonist	TLR4		II	Completed	[82]
	Imiquimod	Agonist	TLR7	NCT04083157	III	Active, not recruiting	
				NCT03307902	II/III	Completed	
				NTR1043		Completed	[85]
	Resiquimod	Agonist	TLR7/8	NCT00175435	I/II	Completed	
	CpG 1018	Agonist	TLR9		I		[87,88]
				NCT00511095	II	Completed	
				NCT00435812	III	Completed	[90]
				NCT02117934	III	Completed	[90,91]
				NCT01005407	III	Completed	[89,90]
				NCT04843852	II	Not yet recruiting	
*HIV*	Poly ICLC	Agonist	TLR3	NCT01127464	I	Completed	
	MPL	Agonist	TLR4	NCT00001042	I	Completed	[92]
	RC-529	Agonist	TLR4	NCT00076037	I	Completed	[93]
	3M-052-AF	Agonist	TLR7/8	NCT04177355	I	Recruiting	
	CpG 1018	Agonist	TLR9	NCT04177355	I	Recruiting	
	CpG7909	Agonist	TLR9	NCT00562939	I/II	Completed	[94]
	Muramyl dipeptide	Agonist	NOD2	NCT00001042	I	Completed	[92]
*HPV*	MPL	Agonist	TLR4	NCT04590521	IV	Not yet recruiting	
				NCT00316706	III	Completed	[95,96]
	Imiquimod	Agonist	TLR7		II		[97]
				NCT01957878	II	Completed	
				NCT00941811	II	Completed	
				NCT00788164	I	Recruiting	
				NCT02689726	I	Terminated	
				NCT00988559	I	Completed	
				ISRCTN32729817	III	Completed	[98,99]
*HSV*	MPL	Agonist	TLR4	NCT00224484	III	Completed	[100]
	CpG 1018 ISS	Agonist	TLR9		III	Completed	
*IAV*	Rintatolimod	Agonist	TLR3	NCT01591473	I/II	Terminated	[101]
	VAX125	Agonist	TLR5	NCT00730457	I	Completed	[102]
				NCT00966238	II	Completed	[103]
	VAX128	Agonist	TLR5	NCT01172054	I	Completed	[103]
	VAX102	Agonist	TLR5	NCT00603811	I	Completed	[104]
				NCT00921973	I	Completed	[105]
	Imiquimod	Agonist	TLR7	NCT01508884	I	Completed	[106]
				NCT03472976	I	Completed	
				NCT02960815	II	Completed	
				NCT02103023	III	Completed	[107]
				NCT04143451	III	Recruiting	
	Resiquimod	Agonist	TLR7/8	NCT01737580	I	Completed

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
