# Peer review of "Anti-Viral Pattern Recognition Receptors as Therapeutic Targets"

_cells, 2021, doi:10.3390/cells10092258_

Round 1

Reviewer 1 Report

This is a short, but well-written review describing PRRs that recognize viruses and current clinical trial efforts to target PRRs in treating disease and vaccine adjuvants. I have a few comments, which I hope will be helpful in improving the review.

Figure 1 is missing a title and legend.

You make it sound as if TLR8 has no role in immunity. There’s a recent paper from Bender et al. in 2020 examining TLR7 vs TLR8 in human cells. There’s also some other literature involving pyogenic infection and viral infection.

On page 4, in discussing the cytosolic DNA sensors, it sounds as if cGAS or AIM2 recognize cyclic dinucleotides. It may help to remove this and mention later, or introduce STING (which actually binds cyclic dinucleotides) earlier.

Bottom of page 4: consider adding a line describing the unmet clinical need (burden of chronic infections, viruses lacking effective vaccines).

Page 5: description of Table 1 is a little uneven, as some trial results reported, but others (e.g. for RO7020531 and TQ-A3334) not described.

Page 8: Consider adding a line describing unmet need in vaccines (lack of efficacy? Durability?) that PRR adjuvants have been developed to address.

It sounds like 2 major approaches to adjuvants have been to include them in the inoculation and as a topical gel. Is that the case? Any evidence for gels being effective?

In the perspectives, some of the focus was surprising. There seems to be a shift to discussing antagonists. A summary of the data and general reflections on efficacy of PRR agonists in disease and vaccine trials are also missing from the conclusions paragraph.

Language issues:

P1 line 40: rather than “These” to start the sentence, consider “PRRs” since previous sentences focused on PAMPs.

Please define “CARD” when it first appears (line 112).

Line 76-77: consider exchanging activate and recruit so recruit goes before “to the site of infection”.

P5 Line 185 capitalize “c”.

Line 342: have been developed and test[ed] – ed missing.

Author Response

Reviewer 1:

This is a short, but well-written review describing PRRs that recognize viruses and current clinical trial efforts to target PRRs in treating disease and vaccine adjuvants. I have a few comments, which I hope will be helpful in improving the review.

We thank the review for their comments and will try to address each of the points below. These are also highlighted in revised file attached.

Figure 1 is missing a title and legend.

We are not sure what is meant here as in the version we have the figure has a title and a legend.

You make it sound as if TLR8 has no role in immunity. There’s a recent paper from Bender et al. in 2020 examining TLR7 vs TLR8 in human cells. There’s also some other literature involving pyogenic infection and viral infection.

We completely agree with this statement and it was not our intention to make it sound like this. We have added a sentence to clarify this.

On page 4, in discussing the cytosolic DNA sensors, it sounds as if cGAS or AIM2 recognize cyclic dinucleotides. It may help to remove this and mention later, or introduce STING (which actually binds cyclic dinucleotides) earlier.

Yes, we agree and have modified the text.

Bottom of page 4: consider adding a line describing the unmet clinical need (burden of chronic infections, viruses lacking effective vaccines).

This has been included.

Page 5: description of Table 1 is a little uneven, as some trial results reported, but others (e.g. for RO7020531 and TQ-A3334) not described.

We agree but unfortunately completed data sets are not available for many of these compounds. While some have publications in journals others are only in conference proceedings or not reported in the literature at all.

Page 8: Consider adding a line describing unmet need in vaccines (lack of efficacy? Durability?) that PRR adjuvants have been developed to address.

A line has now been added to the text.

It sounds like 2 major approaches to adjuvants have been to include them in the inoculation and as a topical gel. Is that the case? Any evidence for gels being effective?

Yes, this appears to be the case for Herpes simplex and human papillomavirus infections.

In the perspectives, some of the focus was surprising. There seems to be a shift to discussing antagonists. A summary of the data and general reflections on efficacy of PRR agonists in disease and vaccine trials are also missing from the conclusions paragraph.

We have modified the text of the conclusions accordingly.

Language issues:

P1 line 40: rather than “These” to start the sentence, consider “PRRs” since previous sentences focused on PAMPs.

Text changed

Please define “CARD” when it first appears (line 112).

Definition included

Line 76-77: consider exchanging activate and recruit so recruit goes before “to the site of infection”.

Text changed

P5 Line 185 capitalize “c”.

Text changed

Line 342: have been developed and test[ed] – ed missing.

Text changed

Reviewer 2 Report

This is an important topic and will be interesting to a wider audience. The text is relatively well written but several typos do exist, those should be omitted.

My main comment is in the fact that it is rather difficult to follow the corresponding text and table 1, also, as the references in-text are numbered and in tables written in full (also in table 2). The references should be numbered also in tables. More importantly, more info about the outcome of the completed trials should be added: a short notion of the safety and of the antiviral efficacy. As such, the tables have less interest. A third table could be also formulated around the most interesting molecules, their safety and efficacy against various targets.

Some other detailed comments:

RLR and NLR abbreviations are defined twice, both differently, see lines 40-41 and 65-66. Define also RIG-I separately as you use that abbreviation throughout the manuscript.

Lines 121-123: give an estimation what is short and long dsRNA  in bps.

Some lettering in Figure 1 is difficult to resolve. The resolution in the final picture should be better.

Lines 155-7: I would say “…unmet clinical need to treat hepatitis…. Infections”.

Line 157: SARS-CoV2 -> SARS-CoV-2

Line 162: rephrase the sentence “Reducing…”.

Author Response

Reviewer 2:

This is an important topic and will be interesting to a wider audience. The text is relatively well written but several typos do exist, those should be omitted.

We thank the review for their comments and will try to address each of the points below. These can also be seen in the revised manuscript attached.

My main comment is in the fact that it is rather difficult to follow the corresponding text and table 1, also, as the references in-text are numbered and in tables written in full (also in table 2). The references should be numbered also in tables. More importantly, more info about the outcome of the completed trials should be added: a short notion of the safety and of the antiviral efficacy. As such, the tables have less interest. A third table could be also formulated around the most interesting molecules, their safety and efficacy against various targets.

Tables have now been updated for consistency. There is still a surprising lack of detailed information on safety from many clinical studies which is why it is not included.

Some other detailed comments:

RLR and NLR abbreviations are defined twice, both differently, see lines 40-41 and 65-66. Define also RIG-I separately as you use that abbreviation throughout the manuscript.

Text has now been updated to avoid repetition.

Lines 121-123: give an estimation what is short and long dsRNA  in bps.

Text has now been included

Some lettering in Figure 1 is difficult to resolve. The resolution in the final picture should be better.

The resolution has been increased to improve the reading of it.

Lines 155-7: I would say “…unmet clinical need to treat hepatitis…. Infections”.

Text has been updated

Line 157: SARS-CoV2 -> SARS-CoV-2

Text has been updated

Line 162: rephrase the sentence “Reducing…”.

Text has been updated